# The Relationship between Lipid Content in Ground Beef Patties with Rate of Discoloration and Lipid Oxidation during Simulated Retail Display

**DOI:** 10.3390/foods10091982

**Published:** 2021-08-25

**Authors:** Yifei Wang, Rubén Domínguez, José M. Lorenzo, Benjamin M. Bohrer

**Affiliations:** 1Department of Animal Sciences, The Ohio State University, Columbus, OH 43210, USA; wang.10408@osu.edu; 2Centro Tecnológico de la Carne de Galicia, Rúa Galicia No. 4, Parque Tecnológico de Galicia, San Cibrao das Viñas, 32900 Ourense, Spain; rubendominguez@ceteca.net (R.D.); jmlorenzo@ceteca.net (J.M.L.); 3Área de Tecnología de los Alimentos, Facultad de Ciencias de Ourense, Universidad de Vigo, 32004 Ourense, Spain

**Keywords:** beef quality, ground beef, fatty acid profile, oxidative stability, color stability

## Abstract

The relationships between the lipid content, lipid oxidation, and discoloration rate of ground beef during a simulated retail display were characterized in this study. A total of 276 batches of ground beef were manufactured with inside rounds and subcutaneous fat from 138 beef carcasses at different targeted levels of lean:fat. There was a total of four different targeted grind levels during the manufacture of the ground beef, and the lipid content for the samples used in this study ranged from 2% to 32% total lipid. Fatty acid composition was determined based on subcutaneous fat, whereas the proximate composition of moisture and total lipids, instrumental color, visual discoloration, and lipid oxidation measured as thiobarbituric acid reactive substances were evaluated on ground beef patties during 7 days of simulated retail at 4 °C display under LED lights. Analysis for the correlation and the creation of linear regression models indicated that lipid content played a more critical role in the discoloration rate compared to lipid oxidation and fatty acid composition. Lipid oxidation could be more reliably predicted by lipid content and instrumental color compared to visual discoloration. Overall, ground beef formulated with greater lipid content is expected to experience greater rates of lipid oxidation and discoloration during retail display.

## 1. Introduction

Consumer purchase intent for ground beef is largely driven by visual appraisal of color, with bright cherry red as the preferred surface color of ground beef when packaged in an aerobic setting [1,2]. Aerobic storage of ground beef is common throughout the world with conventionally over-wrapped trays being a predominant method of display in the retail setting [3,4,5,6]. Aerobic storage can negatively affect the color stability of ground beef through deteriorative effects on the meat protein responsible for color, which is myoglobin. These effects are summarized as the change in the chemical state of myoglobin, which is a transition from oxymyoglobin (red; Fe^2+^) to metmyoglobin (brown, Fe^3+^) [7]. This is an important consideration for the global meat industry since thresholds of approximately 40% metmyoglobin negatively influence consumers’ purchasing decisions [8].

Eating satisfaction and repeat purchase of ground beef products is however largely influenced by flavor and textural attributes [9,10]. Aerobic storage of ground beef can negatively affect the flavor of ground beef through the deteriorative effects caused by lipid oxidation. These effects are summarized as a series of processes where unsaturated fatty acids react with reactive oxygen species, leading to a sequence of secondary reactions, which, in turn, lead to the degradation of lipids and the development of oxidative rancidity [11,12].

While several studies have used predictive modeling to evaluate the relationship between color stability and lipid oxidation [13,14], there does exist some confusion among the relationship between color stability and the lipid oxidation of meat products during extended periods of aerobic storage. As stated by Faustman et al. [15], “Lipid oxidation and myoglobin oxidation in meat lead to off-flavor development and discoloration, respectively. These processes often appear to be linked and the oxidation of one of these leads to the formation of chemical species that can exacerbate oxidation of the other”. The authors [15] go on to discuss that lipid oxidation plays a key role as a facilitator of myoglobin oxidation, and on the other hand, myoglobin plays a key role as a facilitator of lipid oxidation. Similarly, Domínguez et al. [12] also reported that lipid oxidation and heme-protein oxidation occur in a concurrent manner, and each process likely accelerates the other. Nevertheless, applied research that directly investigates the interactive roles of these two oxidative processes is limited.

The objectives of this study were to first evaluate the relationship between the lipid content in ground beef patties with the rate of discoloration and lipid oxidation during simulated retail display, and secondly, to evaluate the relationship between the rate of discoloration and the rate of lipid oxidation of ground beef patties during a simulated retail display period.

It was hypothesized that ground beef manufactured with a greater lipid content could accelerate the rate of the oxidation processes; however, the relationship between the rate of myoglobin state change (i.e., discoloration) and lipid oxidation, while assumed to be linked, may not be strongly related.

## 2. Materials and Methods

The live portion of this research [16,17,18] was approved by the University of Guelph Animal Care Committee (Animal Utilization Protocol #3706). Cattle were received and managed in accordance with the Animal Utilization Protocol, which was approved based on the guidelines and principles of the Canadian Council on Animal Care [19].

### 2.1. Production of Animals and Procurement of Raw Materials

All live animal procedures were previously described by Wang et al. [16] and Dorleku et al. [18]. Briefly, 68 Angus-cross steers were finished on a high-moisture corn, alfalfa silage, and soybean meal diet in 2017 for the Wang et al. [16] study, and 76 Angus-cross steers were finished on a high-moisture corn, alfalfa silage, and soybean meal diet in 2018 for the Dorleku et al. [18] study. Carcass characteristics and meat quality were previously described by Wang et al. [17] and Dorleku et al. [18]. From these two studies, a total of 138 carcasses were used for the current study (63 carcasses from the 2017 study and 75 carcasses from the 2018 study).

Bone-in beef ribs (IMPS#107) and inside rounds (IMPS#168) from the right-side of the carcasses were used in this study [20]. The inside rounds were vacuum packaged at 5-d post mortem and blast frozen (−30 °C). Subcutaneous fat from the beef ribs was collected at 6-d post mortem while maintaining individual carcass identity, vacuum packaged, and blast frozen (−30 °C). Boneless inside rounds and subcutaneous fat were frozen for a period of time ranging from 17 days to 72 days.

### 2.2. Manufacture of Ground Beef Patties

Inside rounds (*semimembranosus*, *adductor*, and other associated muscles) and subcutaneous fat originating from the corresponding bone-in rib were thawed for approximately 5 days at ≤4 °C. All visible fat and connective tissue were trimmed, and pieces were manually cut into approximately 2 cm cubes.

A total of 276 individual batches of ground beef were manufactured for this study over nine different production periods (i.e., independent ground beef processing events) with two different sets of finishing cattle (Wang et al. [16] and Dorleku et al. [18]). The sample size of each production period ranged from 22 to 44 independent ground beef grinds (30.67 samples ± a standard deviation of 6.16 samples). Each production period had an equal number of samples, and two different lean:fat levels (a regular grind and a lean grind) were produced for each carcass, which ultimately generated a wide range of ground beef samples (Figure 1 and Figure 2). Grind 1 and Grind 2 were from beef collected in Wang et al. [16] and consisted of mixing 3.40 kg of cubed lean beef from the inside round with 1.14 kg of cubed subcutaneous fat from the beef ribs (Grind 1) or mixing 4.08 kg of cubed lean beef from the inside round with 0.46 kg of cubed subcutaneous fat from the beef ribs (Grind 2). Grind 3 and Grind 4 were from beef collected in Dorleku et al. [18] and consisted of mixing 3.40 kg of cubed lean beef from the inside round with 1.14 kg of cubed subcutaneous fat from the beef ribs (Grind 3) or grinding 4.54 kg of cubed lean beef from the inside round without additional fat (Grind 4).

Beef was ground using a 3 mm grinding plate attachment equipped to a Sirman Master 90 Y12 meat grinder (Sirman USA, Franklin Park, IL, USA). Each ground beef patty weighed approximately 115 g and was 10 cm in diameter and 1.3 cm thick. From each 4.54 kg grind, six patties were collected and used for this study. Patties were assigned for analyses in the following manner: Patty 1 and Patty 2—proximate composition (moisture and lipid content) and day 0 lipid oxidation using the thiobarbituric acid reactive substances (TBARS) assay; Patty 3, Patty 4, Patty 5, and Patty 6—simulated retail display for the evaluation of color stability and lipid oxidation using the TBARS assay at the conclusion of the display period, which was day 7 of the display period.

### 2.3. Moisture and Lipid Composition

The moisture and lipid concentration of the ground beef samples were determined using modified air drying and the Soxhlet extraction methods, respectively, using the methods previously described by Bohrer et al. [21] and Sivendiran et al. [22]. Duplicate 5 g samples of the ground beef homogenate were weighed onto an aluminum weighing dish and were covered with two (42-mm) #1 Whatman Qualitative filter papers (GE Healthcare Life Sciences, Chicago, IL, USA). Samples were then dried in a forced-air convection drying oven (Fisherbrand Isotemp, Thermo Fisher Scientific, Ottawa, ON, Canada) at 100 °C for 24 h and were then weighed again to determine the moisture content. Dried samples were placed in the Soxhlet extraction apparatus and were washed multiple times for approximately 6–8 h using warm petroleum ether. Washed samples were placed into the 100 °C drying oven for a minimum of 24 h to evaporate any residual petroleum ether and were then weighed to determine lipid lost during the extraction process.

### 2.4. Fatty Acid Profile

A complete description of the methodology used to determine the fatty acid profile, information on the diets that the cattle were fed, and complete lists of fatty acids (including the amount of individual fatty acids expressed in g/100 g tissue) were reported previously in Wang et al. [17] and Dorleku et al. [18].

A sample of subcutaneous fat of approximately 4 cm × 1 cm × 1 cm was collected during fabrication from the posterior end of bone-in ribs. These samples were individually labeled and vacuum packaged before being stored at −30 °C. Samples were then freeze-dried, and the lipids were extracted using 2:1 chloroform:methanol as the solvent and tricosanoic acid as the internal standard. The extraction efficiency of the subcutaneous fat used for fatty acid analysis was 86.70% ± a standard deviation of 4.34% on a dry matter basis. Fatty acid methyl esters were obtained according to methods described by AOCS [23]. A Shimadzu GC-2010 plus, with an AOC-20i/s auto sampler and twin auto injectors, with Shimadzu LabSolutions software was used to quantify the fatty acids. Batches were run on two separate/paired fused silica GC columns (Supelco SP-2560; 100 m × 0.25 mm ID × 0.20 µm film thickness) through split Restek Topaz inlet liners (split ratio of 50). The column temperature was set constant at 140 °C for 5 min, adjusted to 240 °C at 2 °C/min for 50 min, and then held at 240 °C for the final 10 min. The total run time was 65 min/sample. Flame-ionization detection and injector temperature were both set to 250 °C. The fatty acids in samples were identified and quantified by referencing them to known retention times and by using a series of calibration standards (GLC 463, GLC 68E, and 23:0, NuChek Prep., Waterville, MN, USA), respectively. Fatty acid values were reported as g/100g of total tissue, and the average percentage of lipid extracted from the subcutaneous fat samples that were used in this study was 86.70 g/100 g of total tissue on a dry matter basis.

### 2.5. Color Stability

Immediately after the ground beef was manufactured and the patties were formed, two patties and an identification tag were placed on top of a meat soaker pad (Tite-Dri Industries, Boynton Beach, FL, USA) and a Styrofoam tray (Genpak 1005, Genpak, Mississauga, ON, Canada). The patties were then tightly overwrapped with 60-gauge meat wrapping film (Western Plastics, Calhoun, GA, USA) using an Avantco WM-18 single roll film wrapping machine (Webstaurant Store Inc., Lititz, PA, USA). A total of two packages containing two patties (a total of four patties) were created for each ground beef grind. The packages were laid out in a random arrangement onto two multi-level meat display cases in a walk-in refrigerator (temperature was maintained between 2 °C and 4 °C). Each tier of the display case was separated at an equal distance, and each level was illuminated with two 1.22 m long LED tube lights (52 watts, 1850 lumens, 1612.5 to 2152 lux; Acuity Brands Lighting, Conyers, GA, USA). The trays were shuffled once every 24 h so that a more uniform amount of illuminance was applied to all of the samples. Instrumental color and visual surface discoloration (i.e., subjective surface discoloration; % metmyoglobin formation) were evaluated daily until the whole study population reached an average surface discoloration of 60%, which coincided with day 7 of the display period for each of the nine production/display periods. Completing the study at an average surface discoloration aids in the ability to detect meaningful differences among the samples as well as allows for other measurements such as lipid oxidation to be collected on samples at the completion of the simulated retail display.

Instrumental color was evaluated on each day of the display period using a calibrated, handheld Minolta Chroma meter (Konica Minolta Sensing Americas, Inc., Ramsey, NJ, USA) with illuminant D_65_ and 0° viewing angle settings, and 8 mm diameter circular aperture. As per the Commission International de l’Eclairage [24], each measurement using the Chroma meter was reported as *L** (lightness), *a** (redness), and *b** (yellowness). A total of two measurements per patty, four measurements per tray, or eight measurements per experimental unit were collected and then averaged to determine the *L**, *a**, and *b** values for each experimental unit. There were two trained panelists who evaluated visual discoloration (%) on each day of the shelf-life study using the Meat Color Measurement Guidelines outlined by AMSA [25] and the protocols described by Wang et al. [17]. At the end of the study, the trays were vacuum-packaged and stored at −30 °C until further analysis (TBARS assay).

Color change during the display period was assessed using several different calculations. Change over time (Δ_time_) was calculated between day 5 and day 0, between day 6 and day 0, and day 7 and day 0 of the display period for visual discoloration, *L**, *a**, and *b**. Delta E 1976 (Δ*E**_ab_), a measure of the total color difference, was calculated for the difference between the instrumental color on the first and last day of the display period using the Equation (1):(1)$$\Delta E^{*}{}_{\mathrm{ab}}=\sqrt{{\left(\mathrm{Time}\;2\;L^{*}-\;\mathrm{Time}\;1\;L^{*}\right)}^{2}+{\left(\mathrm{Time}\;2\;a^{*}-\;\mathrm{Time}\;1\;a^{*}\right)}^{2}+{\left(\mathrm{Time}\;2\;b^{*}-\;\mathrm{Time}\;1\;b^{*}\right)}^{2}}$$

### 2.6. Thiobarbituric Acid Reactive Substances (TBARS) Assay

In total, two ground beef samples from each experimental unit were used for the TBARS assay. This included one sample that did not undergo the simulated retail display period (day 0) and a random selection of one of the four patties collected at the conclusion of the simulated retail display study (day 7). Day 0 samples were vacuum packaged and blast frozen at −30 °C immediately following the manufacture of the ground beef. The overwrapped trays with the day 7 ground beef patties were vacuum-packaged and blast frozen at −30 °C at the conclusion of the simulated retail display period. All of the ground beef samples remained in the freezer until the TBARS assay was performed.

The TBARS assays were performed using a slightly modified version of the methods described by Leick et al. [26] and Overholt et al. [27]. Duplicate 5 g samples of the ground beef patties were weighed immediately following thawing for 2–3 h in dark refrigeration (≤4 °C). The samples were blended with 1 mL of butylated hydroxytoluene (BHT) and 45.5 mL of 10% trichloroacetic acid in 0.2 M phosphoric acid (TCA/H_3_PO_4_) using a Waring industrial blender (Conair Corporation. Stamford, CT, USA). The blended sample was then filtered through filter paper (No. 1 Whatman; GE Healthcare Life Sciences, Kent, United Kingdom) into two 5 mL duplicates. A total of five mL of thiobarbituric acid (TBA) was then added to one of the two duplicates creating a test sample and a blank sample. The samples were then incubated for 16 h in the dark at room temperature. Samples were then assessed for malondialdehyde (MDA) content using a 96-well plate in a plate reader spectrophotometer (Synergy HT, BioTek Instruments, Inc., Winooski, VT, USA) at a 530 nm wavelength. A standard concentration curve was plotted with 1,1,3,3-tetraethoxypropane (TEP) to determine MDA concentration. Samples were corrected using recovery rate percentages captured using spiked samples, which added 1 mL of 0.2 mg/mL of BHT, 12 mL of TEP, and 32 mL of TCA/H_3_PO_4_. Spiked samples were prepared in duplicate at the same time as the test samples. Therefore, all of the samples were tested in duplicate, and the results were expressed as mg MDA/kg of tissue (i.e., ground beef sample).

### 2.7. Statistical Analyses

The ground beef batch (*n* = 276) served as the experimental unit for all analyses, except for the fatty acid profile, where the carcass (*n* = 138) served as the experimental unit. Summary statistics for all of the variables were computed using PROC MEANS of SAS (version 9.4; SAS Inst., Inc., Cary, NC, USA). The distribution and probability for the variables and their residuals were plotted using the PROC UNIVARIATE of SAS. The assumptions for correlation (i.e., linear relationships exist between variables and the variables are continuous) and the assumptions for regression modeling (i.e., linear relationships exist between the independent and dependent variables, and the residuals are independent, have a constant variance, and are normally distributed) were tested using the distribution plots and the Shapiro–Wilks test for normality generated with the PROC UNIVARIATE of SAS and Levene’s test and the Brown and Forsythe test of homogeneity of variance generated with the PROC GLM of SAS. All statistical assumptions for correlation and regression modeling were met, yet it should be noted that these data were observational in nature, and confounding factors related to diets the animals were fed, fresh meat quality, patty manufacture, and storage conditions during the simulated retail display periods should be recognized.

Pearson correlation coefficients were calculated among all parameters using the PROC CORR of SAS. Correlations were considered weak at |r| < 0.35, moderate at 0.36 ≤ |r| < 0.67, and strong at |r| ≥ 0.68 [28,29]. Coefficients of determination (r^2^) were considered weak at r^2^ < 0.12, moderate at 0.13 ≤ r^2^ < 0.45, and strong at r^2^ ≥ 0.46. Relationships between meaningful variables were further analyzed with simple linear regression using the PROC REG of SAS. The predictions were shown as scatter plots, which were created with Microsoft Excel (Redwood, Washington, DC, USA).

## 3. Results and Discussion

### 3.1. Sample Composition

The study population was characterized with a wide distribution for the ratio between lean and fat, as indicated with the moisture and lipid composition. This was partially expected based on the grinding protocols used in this study, which targeted different levels of lean:fat as well as small batch sizes. Lipid content in the ground beef samples ranged from 2.06% to 32.53%, whereas moisture content varied from 50.68% to 74.30% (Table 1). As expected, the lipid percentage was inversely related to the percentage of moisture, which was consistent with numerous previous research studies [30,31,32,33,34].

Summary statistics for the fatty acid composition of the subcutaneous fat samples used to manufacture ground beef were reported. Total fatty acids, which were calculated by quantification on the basis of the area for the identified peaks using an internal standard, were reported as a sum of saturated fatty acids (SFA), monounsaturated fatty acids (MUFA), and polyunsaturated fatty acids (PUFA) and accounted for 86.70% of the total fat tissue. The content of SFA (42.25 g/100 g of tissue) and MUFA (42.97 g/100 g of tissue) were similar in their values for this population of samples, whereas the PUFA content ranged from 0.64 to 3.41 g/100 g of tissue. Kerth et al. [9] reported the fatty acid profile for 80% lean and 20% fat ground beef samples prepared from the round primal as 48.23% MUFA and 42.99% SFA when expressed as a percentage of total fatty acids. Additionally, Callahan et al. [35] documented greater SFA content (52.17% of total fatty acids) compared to MUFA content (37.59% of total fatty acid content) in ground beef samples formulated to targeted levels of 25% fat. Overall, results from this population of samples were consistent with the typical fatty acid composition expected in beef fat, which is considered as approximately 46% SFA, 51% MUFA, and 3% PUFA when expressed as a percentage of total fatty acids [34,36,37].

### 3.2. Relationship between Lipid Content and Color Stability

Summary statistics for visual discoloration, instrumental color (*L**, *a**, and *b**), total color difference (Δ*E**_ab_), and TBARS over the 7-day retail display period are presented in Table 2. The relationship between the lipid content and the ground beef color stability determined by Pearson’s correlation coefficients are summarized in Table 3. A moderate positive correlation (r = 0.53; *p* < 0.01) was observed between the lipid content and the change in the visual discoloration from day 0 to day 7 of the retail display period (visual discoloration Δ_Day-7–Day-0_). This indicated that the beef patties formulated with greater fat experienced a greater rate of visual discoloration during the retail display period. Similarly, Shivas et al. [38] reported that ground beef samples formulated with 25% fat underwent visual discoloration at a faster rate than counterpart samples formulated with 20% fat after 5 days of retail display. Likewise, Callahan et al. [35] observed a more rapid visual discoloration in ground beef samples formulated with 25% fat than 5% fat counterpart samples after 7 days of simulated retail display. In partial agreement, an interaction between fat level and retail display time was observed for visual discoloration by Pietrasik et al. [39], who reported that lean (15% fat) beef patties experienced greater visual discoloration on day 1 and day 2 of display, whereas regular (30% fat) beef patties exhibited greater visual discoloration on day 3 of display. In contrast to our results, Garner et al. [33] documented Premium Choice (upper two-thirds of USDA Choice) ground beef with greater fat content demonstrated less visual discoloration than ground beef formulated with USDA Select subprimals after 3 days of retail display.

A moderate positive correlation (r = 0.46; *p* < 0.01) was observed between the lipid content and the initial lightness (*L** Day-0 scores), suggesting that beef patties formulated with greater fat demonstrated greater lightness at the beginning of retail display. The increase of the *L** value could be attributed to the increase in the light-scattering properties associated with fat particles [40,41]. In agreement, numerous previous investigations [9,33,35,39,42,43,44] have documented that an increase in fat content generated a lighter visual appearance of ground beef. In addition, there was a weak positive correlation (r = 0.32; *p* < 0.01) observed between the lipid content and the change in the *L** value during retail display (*L** Δ_Day-7–Day-0_). This finding revealed that lipid content may not be the only contributor to the change in lightness observed in beef patties during display and that several other factors are also involved, such as lipid oxidation [9,45] and muscle pH [46]. High ultimate pH could improve the water holding capacity and increase the content of water-soluble myoglobin in the meat system. Additionally, high pH could cause muscle fibers to swell and tightly pack together, preventing the diffusion of oxygen and the absorption of light, leading to a darker appearance of meat [46].

There was a strong positive correlation (r = 0.63; *p* < 0.01) observed between lipid content and initial redness (*a** Day 0 scores). This was unexpected, as greater lipid content in ground meat samples is generally related to a less red appearance. However, a strong negative correlation (r = −0.70; *p* < 0.01) was observed between lipid content and the change in redness from day 0 to day 7 of the display period (*a** Δ_Day-7–Day-0_). These findings suggested that while beef patties formulated with greater fat demonstrated greater redness at day 0, a sharper decrease in redness was observed in beef patties formulated with greater fat during the retail display period, compared to the lower-fat counterparts. In support, Houben et al. [47] reported increased fat content contributed to a greater initial *a** value for minced beef and greater discoloration after 5 days of retail display. A similar pattern was observed by Garner et al. [33], who reported that ground beef made from chuck roll cuts and Premium Choice (upper two-thirds of USDA Choice) subprimals that contained a greater percentage of fat than those manufactured from knuckle and USDA Select subprimals demonstrated greater initial redness and a faster rate of discoloration after display for 3 days. Likewise, Cooper et al. [48] observed that ground beef formulated with 25% fat demonstrated lower *a** values than counterpart samples formulated with 5% fat during 7 days of retail display. On the contrary, Pietrasik et al. [39] and Callahan et al. [35] reported a greater initial *a** value in beef patties formulated with lower fat content. In addition, Reynolds et al. [49] documented that ground beef samples formulated with 5% fat demonstrated a greater *a** value than the 25% fat counterpart samples after 7 days of retail display. The different light sources utilized by Reynolds et al. [49], which could influence meat color stability and lipid oxidative stability [48,50,51,52], might have contributed to the observed variation in the *a** value.

A strong positive correlation (r = 0.73; *p* < 0.01) was observed between lipid content and initial yellowness (*b** Day-0 scores), suggesting beef patties formulated with greater fat had a greater *b** value. This observation was expected based on numerous previous research studies [32,33,35,39,48], which have documented greater initial yellowness for higher fat ground beef formulations. Similar results also have been reported in pork sausages [40,53]. A moderate negative correlation (r = −0.36; *p* < 0.01) was observed between lipid content and the change in yellowness from day 0 to day 7 (*b** Δ_Day-7–Day-0_). This indicated that beef patties formulated with greater fat had less change in yellowness during retail display. In general, high lipid content favored an increase in initial yellowness and slowed down the change in yellowness during retail display. In contrast, Ismail et al. [54] reported a sharper decline of yellowness in ground beef formulated with 20% fat compared to 10% fat counterpart samples during 7 days of retail display.

The calculated total color difference (Δ*E**_ab_) of the ground beef samples during the display period, which accounts for combined changes in *L**, *a**, and *b*,* can be used as an indicator of discoloration during a retail display period [55,56]. There was a strong positive correlation (r = 0.68; *p* < 0.01) observed between lipid content and total color difference (Δ*E**_ab_) from day 0 to day 7 of the display period. This indicated that beef patties formulated with greater fat underwent greater total color difference during retail display. Likewise, Bhattacharya et al. [57] reported that the total color difference of ground beef patties was affected by fat content and storage time. However, conflicting results have been documented by Liu et al. [58], who reported that patties formulated with 20% fat exhibited lower Δ*E**_ab_ compared to counterpart samples formulated with 10% fat after 10 days of display. This indicated that the patties with a greater fat level demonstrated improved color stability.Linear regression was used to generate prediction equations between meaningful parameters (Figure 3). It was determined that the lipid percentage in beef patties was a strong predictor for *a** Δ_Day-7–Day-0_ (r^2^ = 0.49; Figure 3C) and Δ*E**_ab_ (r^2^ = 0.46; Figure 3E). This indicates that changes in redness and changes in total color during a retail display period could be reliably predicted using the fat content of beef patties. Moreover, the lipid percentage in beef patties was a moderate predictor for visual discoloration % Δ_Day-7–Day-0_ (r^2^ = 0.28; Figure 3A) and *b** Δ_Day-7–Day-0_ (r^2^ = 0.13; Figure 3D), indicating that visual discoloration and the change in yellowness during retail display could be partially attributed to the lipid content of the beef patties. Furthermore, *L** Δ_Day-7–Day-0_ was weakly predicted with lipid percentage in beef patties (r^2^ = 0.10; Figure 3B), meaning that lipid content plays a minimum role in predicting the change in the lightness of beef patties during retail display.

In post mortem muscle, unsaturated fatty acids located near the cellular membranes of the myofiber and its constituents of subcellular organelles could potentially interact with the heme iron component of myoglobin, which could accelerate both lipid oxidation and myoglobin oxidation [59]. Nevertheless, results from the current study suggested that fatty acid composition in subcutaneous fat and the discoloration rates of ground beef were weakly correlated (|r| < 0.35; Table 4). Among all of the correlations, the strongest correlations were observed between PUFA:MUFA and the change in redness (*a** Δ_Day-5–Day-0_; r = −0.23; *p* < 0.01). This indicated that beef containing greater levels of PUFA and lower levels of MUFA is expected to experience a greater rate of decline in redness. Nevertheless, Gatellier et al. [60] observed that intramuscular fat of *longissimus dorsi* muscle from pasture-diet finished beef demonstrated a greater PUFA content, and a similar MUFA content experienced a slower rate of decrease in redness during 6 days of retail display compared to mixed-diet finished beef. PUFA, which are mostly found in the phospholipid fraction of muscle and adipose tissue [61], are the major substrates for lipid peroxidation in meat [60], which could lead to the greater discoloration. Moreover, Ponnampalam et al. [62] suggested that PUFA could explain 42.4% of the decrease of redness in lamb during retail display. Additionally, it was reported that the relationship between PUFA and the redness of lamb might be mediated through the interactions of heme iron and vitamin E [62]. Furthermore, the elevated level of n-6 PUFA might compromise meat color stability through oxidation, which generates 4-hydroxynoenal (HNE), a highly reactive electrophilic molecule [63,64]. 4-hydroxynoenal has been documented to compromise beef myoglobin redox stability through Michael adduction both in vitro [65,66,67] and in vivo [68] and ultimately leads to meat discoloration. Overall, results from the current study suggest that the color stability of ground beef may not be directly influenced by fatty acid composition.

### 3.3. Relationship between Lipid Content and Lipid Oxidation

The relationship between lipid content and lipid oxidation for ground beef determined by Pearson’s correlation coefficients were summarized in Table 5. There was not a significant correlation (r = 0.09; *p* = 0.17) between lipid content and initial lipid oxidation (Day-0 TBARS). Nevertheless, a moderate positive correlation (r = 0.46; *p* < 0.01) was observed between lipid content and lipid oxidation on day 7 (Day-7 TBARS). These findings indicated that increased fat content did not influence lipid oxidation at the beginning of retail display; however, it contributed to greater TBARS values in ground beef samples after 7 days of retail display. In agreement, numerous investigations [33,42,44,48,54] observed similar TBARS values at the beginning of retail display, regardless of fat content in ground beef. Similar to the results of this study, Liu et al. [58] documented that ground beef samples formulated with 20% fat experienced greater lipid oxidation compared to the counterpart samples formulated with 10% fat after 15 days of retail display. In addition, Raines et al. [43] reported greater lipid oxidation in ground beef formulated with greater fat content in a high oxygen modified atmosphere for 4 days of retail display. While Martin et al. [32], Lavieri and Williams [44], and Callahan et al. [35] observed an increase in the TBARS values upon retail display, no significant differences among the lean formulations were noted on day 7 of retail display. In contrast to the current study results, Cooper et al. [48] reported that ground beef formulated with 5% fat demonstrated greater oxidation than counterpart samples formulated with 25% fat on day 7 of retail display. Contradicting results of the current study were also reported by Houben et al. [47], who observed lean minced beef samples that had been formulated with 0.9% fat exhibited greater lipid oxidation than counterpart samples formulated with 19.9% of fat after 7 days of display.

There is a long-standing debate among meat scientists regarding the influence of fat content on lipid oxidation. In this regard, it is well known that the phospholipid components of meat samples have an important role in lipid oxidation due to their high composition of unsaturated fatty acids and their arrangement in membranes, which facilitates the propagation step of oxidation reactions [12]. Moreover, due to this aspect, lean meat (with high amounts of phospholipids) is very susceptible to lipid oxidation [69], which partially explains the findings described by Houben et al. [47]. Thus, it is plausible that the fatty acid composition and the total fat content are important factors in the development of lipid oxidation since the proportions of triglycerides and phospholipids play a vital role in the development and propagation of oxidative reactions [12]. However, and after considering that fat is the substrate for the development of lipid oxidation, meat products formulated with a higher fat content should have greater substrate availability that can undergo oxidative processes. Therefore, the complexity of the reactions and multiple other factors involved make it difficult to understand the role that the amount of fat has on oxidative processes.

In the current study, a moderate positive correlation (r = 0.44; *p* < 0.01) was observed between the lipid content and the lipid oxidation rate (TBARS Δ_Day-7–Day-0_). This indicated that ground beef formulated with greater fat experienced greater rates of lipid oxidation during retail display. In order to compare these results with previous investigations, the rate of lipid oxidation in the following discussion was calculated as the differences (in absolute value) between the initial TBARS values and the final TBARS values during retail display, using the TBARS results in previous publications. In agreement with the findings from the current study, results from Ismail et al. [54] suggested that ground beef samples formulated with 20% fat demonstrated a greater increase in the TBARS values compared to 10% and 15% fat counterpart samples after 10 days of retail display. Similar results were documented by Martin et al. [33], who reported that ground beef samples formulated with 27% fat underwent lipid oxidation at a greater rate compared to 19% fat and 9% fat counterpart samples during 28 days of dark, refrigerated storage. By contrast, Houben et al. [47], Raines et al. [43], Lavieri and Williams [44], and Cooper et al. [48] reported that ground beef formulated with a greater fat content demonstrated slower rates of increase in the TBARS values when measured before and following retail display. In addition to the aforementioned factors that could affect these contradicting results, different oxidation rates might be associated with the variation in the fatty acid composition of beef fat. Furthermore, it was determined by the linear regression analysis that lipid percentage was a moderate predictor for TBARS Δ_Day-7–Day-0_ (r^2^ = 0.19; Figure 3F). This suggested that the rate of lipid oxidation during retail display could be partially attributed to the total lipid content in ground beef.

The correlations between fatty acids and the rate of lipid oxidation in ground beef were generally weakly correlated (|r| < 0.35; Table 6). The only exception was the correlation between PUFA:MUFA and TBARS Δ_Day-7–Day-0_, which was on the cusp of being moderately correlated (r = 0.31; *p* < 0.01). This indicated that the greater ratio of PUFA:MUFA could lead to greater change in TBARS values during retail display. In other words, beef containing higher levels of PUFA and lower levels of MUFA tend to experience lipid oxidation at a faster rate. This observation was expected from a theoretical standpoint since the energy required to remove a hydrogen atom from methylene carbon is lower than the energy required to remove a hydrogen atom from methyl carbon; therefore, PUFA are oxidized at a faster rate compared to MUFA [70]. Interestingly, findings from the current study indicated that the ratio between PUFA and MUFA in ground beef might play a more critical role than the total PUFA or MUFA content in accelerating lipid oxidation.

Lipid content and fatty acid composition are considered as the main factors that influence lipid oxidation [12]. However, there has been extensive discussion among meat scientists regarding which factor plays a more meaningful role in lipid oxidation. Previous researchers [12,71,72] suggested that fatty acid composition is more important than lipid content in accelerating lipid oxidation in whole muscle cuts. The current study indicated that lipid content is a more critical contributor to lipid oxidation compared to fatty acid composition in ground beef. The population of samples in this study could have contributed to this result—the lipid content of the samples in this study ranged from 2.06% to 32.53%, while PUFA:MUFA ranged from 0.01 to 0.07. Ground beef tends to experience greater lipid oxidation than whole muscle cuts due to the grinding process, which incorporates oxygen [15] and potentially compromises the integrity of cellular and subcellular membranes, increasing the chance of an interaction between unsaturated fatty acid and iron-containing proteins [59]. Therefore, whether fatty acid composition or lipid content has a greater impact on lipid oxidation may depend on the form of the meat products.

### 3.4. Relationship between Lipid Oxidation and Color Stability

Lipid oxidation and myoglobin oxidation are observed to occur in a concurrent manner in muscle foods [12,15,73,74]. During lipid oxidation, unsaturated fatty acids react with reactive oxygen species, generating a wide range of primary and secondary products, such as aldehydes and ketones [15,75,76]. 4-hydroxynoenal, a well-documented secondary product of n-6 PUFA oxidation in meat can adduct to histidine, lysine, and cysteine residues through alkylation and can therefore induce myoglobin oxidation [66,67,68]. When the central iron atom within the heme group of myoglobin is oxidized, the ferrous heme iron is converted to its ferric form, resulting in brownish metmyoglobin, which is responsible for meat discoloration [15]. As a result, lipid oxidation is believed to enhance meat discoloration [15]. Moreover, the reactive intermediates generated through myoglobin oxidation can act as prooxidants and further enhance lipid oxidation [15,74]. Additionally, HNE alkylation could compromise myoglobin tertiary structure [68], leading to heme exposure and even the release of iron, which could, in turn, catalyze lipid oxidation [15]. Therefore, myoglobin oxidation and lipid oxidation are considered to facilitate each other.

Pearson correlation coefficients between lipid oxidation and color stability are shown in Table 7. The rate of visual discoloration and change in instrumental color during the retail display period of ground beef was not significantly correlated (*p* > 0.05) with initial lipid oxidation (Day-0 TBARS). However, the discoloration rate of ground beef (measured visually and instrumentally) was significantly correlated (*p* < 0.05) with the lipid oxidation rate (TBARS Δ_Day-7–Day-0_) and the lipid oxidation level at the end of retail display (Day-7 TBARS).

Moderate positive correlations (r ≥ 0.35; *p* < 0.01) were observed between the rate of lipid oxidation (TBARS Δ_Day-7–Day-0_) and visual discoloration during retail display (visual discoloration Δ_Day-7–Day-0_) and between final lipid oxidation level (Day-7 TBARS) and the rate of visual discoloration (visual discoloration Δ_Day-7–Day-0_). This suggested that ground beef that experienced greater lipid oxidation during the retail display period demonstrated a greater rate of visual discoloration. Similarly, Shivas et al. [38] reported that visual discoloration and lipid oxidation were linearly related when they observed lower levels of visual discoloration and lower levels of lipid oxidation in ground beef samples treated with 10% ascorbic acid compared to non-treated counterpart samples after 10 days of retail display. Moreover, Garner et al. [33] documented that ground beef prepared with knuckle subprimals, which demonstrated a greater level of lipid oxidation than those prepared with chuck roll subprimals, experienced greater visual discoloration after 3 days of retail display. Nevertheless, Pietrasik et al. [39] reported that although ground beef formulated with 30% fat underwent greater lipid oxidation than the 15% fat counterpart samples after 3 days of retail display, no significant differences in visual discoloration were observed among the two formulations. The differences in the results might be attributed to the shorter retail display period conducted by Pietrasik et al. [39].

The observed change of lightness during the retail display period (*L** Δ_Day-7–Day-0_) was moderately correlated with both lipid oxidation rate (TBARS Δ_Day-7– Day-0_; r = 0.50; *p* < 0.01) and the final level of lipid oxidation (Day-7 TBARS; r = 0.44; *p* < 0.01). These findings indicated that ground beef that had a faster rate and greater level of lipid oxidation experienced a greater change in lightness during retail display. In contrast, Cooper et al. [48] observed that ground beef formulated with 5% fat underwent a greater rate of lipid oxidation during 7 days of retail display and exhibited a greater decline in lightness compared to those formulated with 25% fat. Conflicting results were also reported by Kerth et al. [9], who documented a greater increase for the *L** value in ground beef prepared with chuck fat compared to those prepared with round fat after 5 days of retail display, however, no significant differences in lipid oxidation were observed among subcutaneous fat sources. While an increase in the *L** value over 7 days of retail display was observed in the current study, several other investigations [5,33,43,44,48,54,77] observed decreased lightness in ground beef during retail display. Overall, ground beef was expected to experience a minor change in the *L** value during lipid oxidation since the lightness of meat is mainly determined by pigment content, which remained stable during retail display [78].

There was a moderate negative correlation (r = −0.55; *p* < 0.01) between the rate of lipid oxidation (TBARS Δ_Day-7–Day-0_) and the change in redness (*a** Δ_Day-7–Day-0_). Additionally, a moderate negative correlation (r = −0.55; *p* < 0.01) was also observed between the levels of lipid oxidation on day 7 (Day-7 TBARS) and changes in redness (*a** Δ_Day-7–Day-0_). These results revealed that the ground beef, which demonstrated a faster lipid oxidation rate and a greater level of lipid oxidation at the end of retail display, experienced a sharper decline in redness. In agreement with the findings reported in the current study, Suman et al. [5] reported that aerobic packaged ground beef exhibited a greater lipid oxidation rate and experienced a greater loss of redness than those packaged in a high oxygen modified atmosphere packaging system after 3 days of dark, refrigerated storage. Moreover, greater lipid and redness stability were observed in chitosan-treated ground beef samples compared to non-treated ground beef samples during 3 days of refrigerated storage [5]. Likewise, Lee et al. [79] reported that the incorporation of antioxidants including sodium citrate, sodium erythorbate, and rosemary extract in *n*-3 oil fortified ground beef delayed both lipid oxidation and the decline of redness. Additionally, the observations from the current study were supported by several investigations [43,44,48], which reported that ground beef samples formulated with less fat content demonstrated a faster rate of lipid oxidation and discoloration compared to counterpart samples formulated with greater fat content during retail display. The lower fat content in ground beef should increase the amount of muscle tissue in the ground beef product and should therefore increase the amount of muscle color pigment available for discoloration reactions to occur [48].

A moderate negative correlation (r = −0.38; *p* < 0.01) was observed between the rate of lipid oxidation (TBARS Δ_Day-7–Day-0_) and the change in yellowness (*b** Δ_Day-7–Day-0_), indicating that the greater lipid oxidation rate in ground beef favored a greater loss of yellowness during retail display. In support, Martin et al. [32] and Lavieri and Williams [44] reported that ground beef samples formulated with greater fat and exhibiting greater lipid oxidation rates demonstrated a greater decline in yellowness compared to the lower fat counterpart samples. In contrast, Ismail et al. [54] documented a greater lipid oxidation rate and lower rates of decrease for *b** values in ground beef formulated with greater fat content during retail display. Overall, results from the current study were in agreement with the observation of Salueña et al. [78], who reported that the changes in *b** values of meat were less pronounced than in *a** values during retail display.

Total color difference was considered as a more reliable measurement than the visual inspection for meat discoloration [78]. Similar to other instrumental color attributes, a moderate positive correlation (r = 0.57; *p* < 0.01) was observed between total color difference (Δ*E**_ab_) and lipid oxidation rate during the retail display (TBARS Δ_Day-7–Day-0_). In addition, there was a moderate positive correlation (r = 0.56; *p* < 0.01) between total color difference (Δ*E**_ab_) and final lipid oxidation level (Day-7 TBARS). These results indicated that ground beef exhibited greater lipid oxidation rate, a greater final level of lipid oxidation, and experienced greater total discoloration during retail display. In agreement, Zamuz et al. [80] documented that ground beef incorporated with either chestnut extracts or BHT, which experienced lower lipid oxidation rate, demonstrated lower level of Δ*E**_ab_ compared to control ground beef during 18 days of retail display. Likewise, ground beef treated with black rice water extract decreased the rate of lipid oxidation and total color difference over 6 days of retail display [81]. A similar observation was also reported by Hashemi Gahruie et al. [82], who documented that the incorporation of antioxidants, including thyme, cinnamon, rosemary extracts, and BHT delayed both lipid oxidation and total discoloration in beef burgers frozen at −18 °C for 60 days.

Linear regression results (Figure 4) indicated that visual discoloration rate was a moderate predictor for lipid oxidation rate (TBARS Δ_Day-7–Day-0_; r^2^ = 0.15; Figure 4A); however, this prediction ability was considerably lower than instrumental means of color measurement. This finding was expected–while human visual systems operate well to differentiate colors presented simultaneously, human visual systems have a poor capacity to memorize color differences and notice meat discoloration during retail display [78]. *L** Δ_Day-7–Day-0_ (r^2^ = 0.24; Figure 4B), *a** Δ_Day-7–Day-0_ (r^2^ = 0.33; Figure 4C), *b** Δ_Day-7–Day-0_ (r^2^ = 0.15; Figure 4D), and Δ*E**_ab_ (r^2^ = 0.34; Figure 4E) were moderate predictors for the lipid oxidation rate (TBARS Δ_Day-7–Day-0_). Results from the current study revealed that the lipid oxidation rate could be partially attributed to the meat discoloration rate during retail display.

Overall, the current study suggested that the lipid oxidation rate and the discoloration rate are moderately correlated, indicating that there might be other independent factors contributing to meat discoloration and lipid oxidation. For example, mitochondrial function could impact meat color stability through the regulation of the myoglobin redox state [59,75,83]. In post mortem muscle, mitochondria compete with myoglobin for oxygen; the greater amount of oxygen utilized for mitochondrial oxygen consumption results in a less amount of oxygen that is available for myoglobin, which could hinder the development of the desirable cherry red color of beef [84]. Nonetheless, mitochondrial oxygen consumption could enhance metmyoglobin reduction by transferring available electrons to metmyoglobin [85] and could potentially improve meat color stability. Additionally, mitochondrial function could be limited by HNE in vitro [86]. In general, meat color might be determined by the equilibrium of lipid oxidation, myoglobin oxidation, and mitochondria activity. Moreover, several recent investigations [68,87,88] have discovered myoglobin post-translational modifications, especially phosphorylation, which could influence myoglobin redox stability and compromise meat color stability. However, the relationship between lipid oxidation and phosphorylation remains unknown and warrants greater research effort.

## 4. Conclusions

Fatty acid composition was not the main contributor to the lipid oxidation and discoloration of ground beef. There was a positive relationship between PUFA:MUFA and the change in TBARS during the display period, yet the strength of this relationship was only on the cusp of being moderately correlated. In comparison to lipid oxidation rate, the total lipid content was more closely associated with the discoloration rate, especially regarding the decline of redness and total color difference. In addition, lipid content could be used to moderately predict lipid oxidation rate (as measured with TBARS) and could reliably predict the decline of redness and the total color difference in ground beef during retail display, indicating ground beef formulated with a greater lipid content is expected to experience greater rates of lipid oxidation and discoloration. While the rate of lipid oxidation could be moderately predicted using instrumental color attributes measured during retail display, the rate of lipid oxidation could not be reliably predicted by visual discoloration.

## Figures and Tables

**Figure 1 foods-10-01982-f001:**
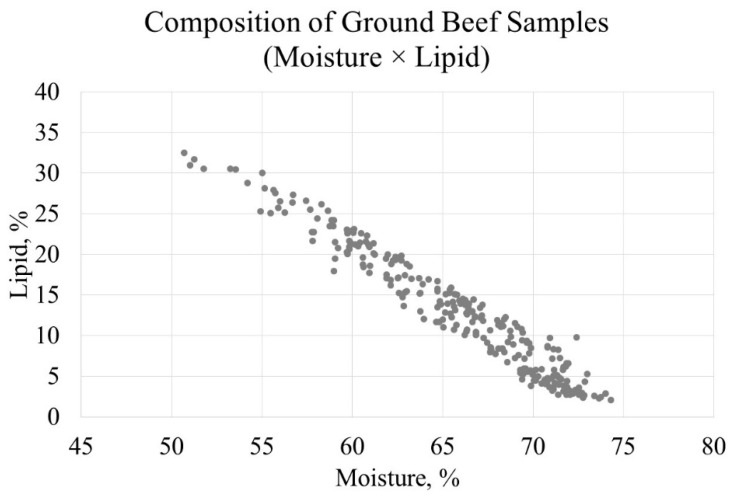
Scatterplot representing composition (moisture × lipid) of ground beef samples used in this study (*n* = 266).

**Figure 2 foods-10-01982-f002:**
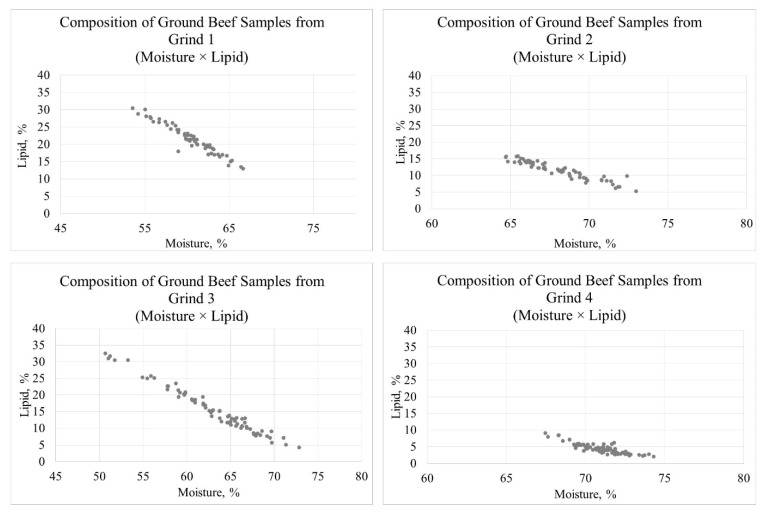
Scatterplot representing composition (moisture × lipid) of ground beef samples used in this study broken down into the four different grinds; Grind 1: average moisture = 60.57 ± stdev. 3.04% and average lipid 21.36 ± stdev. 4.20%; Grind 2: average moisture = 68.11 ± stdev. 2.21% and average lipid 11.54 ± stdev. 2.70%; Grind 3: average moisture = 63.10 ± stdev. 5.04% and average lipid 15.45 ± stdev. 6.81%; Grind 4: average moisture = 71.03 ± stdev. 1.45% and average lipid 4.44 ± stdev. 1.53%.

**Figure 3 foods-10-01982-f003:**
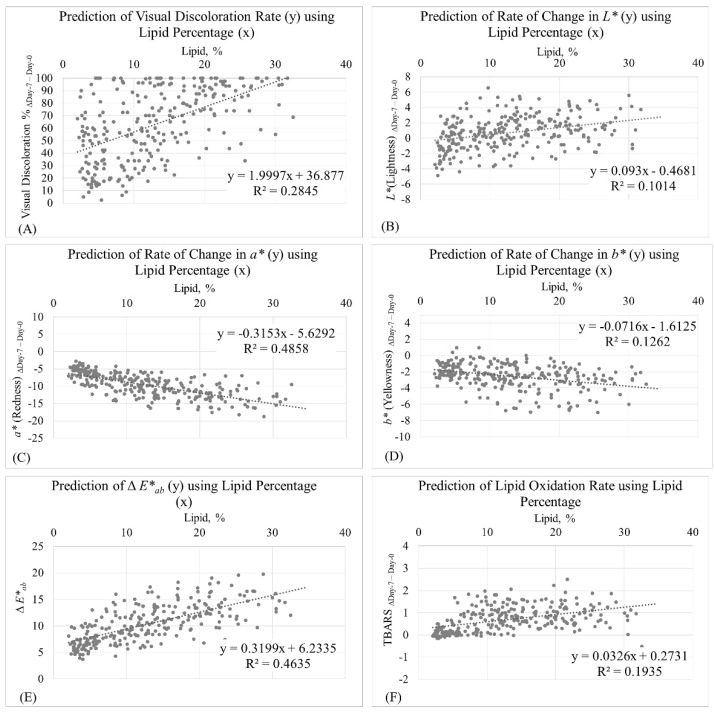
Prediction of visual discoloration rate (Δ_Day-7–Day-0_) using the lipid percentage of ground beef as the independent variable (**A**); prediction of change in instrumental *L** (Δ_Day-7–Day-0_) using lipid percentage of ground beef as the independent variable (**B**); prediction of change in instrumental *a** (Δ_Day-7–Day-0_) using lipid percentage of ground beef as the independent variable (**C**); prediction of change in instrumental *b** (Δ_Day-7–Day-0_) using lipid percentage of ground beef as the independent variable (**D**); prediction of ΔE*_ab_ using lipid percentage of ground beef as the independent variable (**E**); prediction of lipid oxidation rate (TBARS Δ_Day-7–Day-0_) using lipid percentage of ground beef as the independent variable (**F**).

**Figure 4 foods-10-01982-f004:**
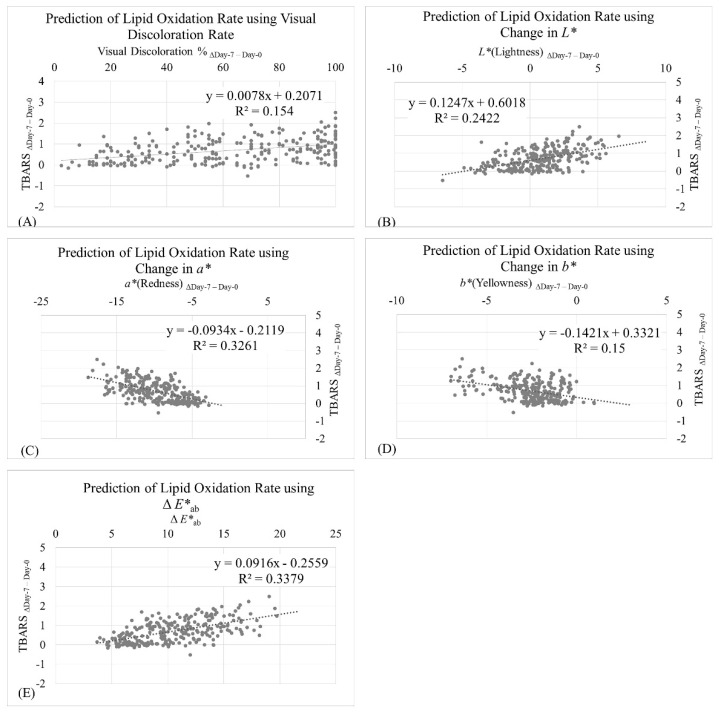
Prediction of lipid oxidation rate (TBARS Δ_Day-7–Day-0_) using visual discoloration rate (Δ_Day-7–Day-0_) (**A**); change in instrumental *L** (Δ_Day-7–Day-0_) (**B**); change in instrumental *a** (Δ_Day-7–Day-0_) (**C**); change in instrumental *b** (Δ_Day-7–Day-0_) (**D**); and change in instrumental Δ*E**_ab_ (**E**) as independent variables.

**Table 1 foods-10-01982-t001:** Summary statistics for the composition of ground beef and the fatty acid profile of the subcutaneous fat tissue used to manufacture ground beef.

Variable	N	Mean	Std. Dev.	Minimum	Maximum	Coeff. of Variation, %
Moisture, %	266	65.83	5.27	50.68	74.30	8.00
Lipid, %	266	12.87	7.57	2.06	32.53	58.84
Fatty acid profile ^1^, g/100 g of tissue						
Total FA in tissue	138	86.70	4.34	73.68	95.11	5.00
Total SFA ^2^	138	42.25	3.70	32.07	54.06	8.75
Total MUFA ^3^	138	42.97	3.75	28.23	58.11	8.73
Total PUFA ^4^	138	1.48	0.36	0.65	3.41	24.67
MUFA:SFA ^5^	138	0.952	0.149	0.522	1.665	15.72
PUFA:SFA ^6^	138	0.033	0.009	0.014	0.088	28.93
PUFA:MUFA ^7^	138	0.032	0.008	0.014	0.074	25.42
n-6 PUFA ^8^	138	1.095	0.320	0.397	2.917	29.18
n-3 PUFA ^9^	138	0.382	0.124	0.183	0.882	32.51
n-6:n-3 ratio ^10^	138	2.875	0.989	0.696	5.932	34.41

^1^ Fatty acid profile was determined on subcutaneous fat from each carcass (two ground beef grinds were created from each carcass). ^2^ Total saturated fatty acids = C14:0 + C15:0 + C16:0 + C17:0 + C18:0 + C19:0 + C20:0 + C22:0 + C24:0. ^3^ Total monounsaturated fatty acids = C14:1 n-5 + C15:1 n-5 + C16:1 n-7 + C17:1 n-7 + C18:1 n-9 + C19:1 n-9 + C20:1 n-9. ^4^ Total polyunsaturated fatty acids = C18:2 n-6 + C18:3 n-3 + C18:4 n-3 + C20:2 n-6 + C20:3 n-3 + C20:4 n-6 + C20:5 n-3 + C22:2 n-6 + C22:3 n-3 + C22:4 n-6 + C22:5 n-3 + C22:6 n-3. ^5^ MUFA:SFA = total monounsaturated fatty acids ÷ total saturated fatty acids. ^6^ PUFA:SFA = total polyunsaturated fatty acids ÷ total saturated fatty acids. ^7^ PUFA:MUFA = total polyunsaturated fatty acids ÷ total monounsaturated fatty acids. ^8^ Total n-6 polyunsaturated fatty acids = C18:2 n-6 + C18:3 n-6 + C20:2 n-6 + C20:3 n-6 + C20:4 n-6 + C22:2 n-6 + C22:4 n-6. ^9^ Total n-3 polyunsaturated fatty acids = C18:3 n-3 + C18:4 n-3 + C20:3 n-3 + C20:5 n-3 + C22:5 n-3 + C22:6 n-3. ^10^ n-6:n-3 ratio = n-6 fatty polyunsaturated fatty acids ÷ n-3 polyunsaturated fatty acids.

**Table 2 foods-10-01982-t002:** Summary statistics for color and lipid oxidation during a 7-day simulated retail display.

Variable	N	Mean	Std. Dev.	Minimum	Maximum	Coeff. of Variation, %
Visual discoloration ^1^, %						
Day 0 Scores	276	0.00	0.00	0.00	0.00	-
Day 5 Scores (Δ_Day-5–Day-0_)	276	28.01	25.17	0.00	100.00	89.92
Day 6 Scores (Δ_Day-6–Day-0_)	276	43.98	29.26	0.00	100.00	66.54
Day 7 Scores (Δ_Day-7–Day-0_)	276	62.73	28.35	2.50	100.00	45.20
*L** (lightness)^2^						
Day 0 Scores	276	48.89	3.32	40.66	58.70	6.80
Day 5 Scores (Δ_Day-5–Day-0_)	276	49.53	3.07	43.20	58.29	6.21
Day 6 Scores (Δ_Day-6–Day-0_)	276	49.34	3.15	42.32	58.96	6.38
Day 7 Scores (Δ_Day-7–Day-0_)	276	49.67	3.22	42.52	58.54	6.49
Δ_Day-5–Day-0_	276	0.65	1.49	−3.14	4.78	230.23
Δ_Day-6–Day-0_	276	0.46	1.98	−4.94	5.87	434.16
Δ_Day-7–Day-0_	276	0.78	2.21	−6.46	6.55	283.34
*a** (redness)^2^						
Day 0 Scores	276	20.76	2.72	14.68	26.71	13.09
Day 5 Scores (Δ_Day-5–Day-0_)	276	14.40	2.07	8.12	19.35	14.40
Day 6 Scores (Δ_Day-6–Day-0_)	276	12.71	2.15	7.21	18.54	16.89
Day 7 Scores (Δ_Day-7–Day-0_)	276	11.05	2.20	6.25	16.47	19.88
Δ_Day-5–Day-0_	276	−6.36	2.47	−15.65	−1.69	−38.81
Δ_Day-6–Day-0_	276	−8.04	3.13	−17.90	−1.97	−38.94
Δ_Day-7–Day-0_	276	−9.70	3.41	−18.76	−2.79	−35.09
*b** (yellowness) ^2^						
Day 0 Scores	276	9.98	2.44	5.23	16.29	24.48
Day 5 Scores (Δ_Day-5–Day-0_)	276	7.71	1.64	4.32	11.22	21.28
Day 6 Scores (Δ_Day-6–Day-0_)	276	7.57	1.67	3.61	11.60	22.07
Day 7 Scores (Δ_Day-7–Day-0_)	276	7.46	1.64	2.91	11.06	22.00
Δ_Day-5–Day-0_	276	−2.26	1.46	−6.95	−0.21	−64.47
Δ_Day-6–Day-0_	276	−2.41	1.65	−7.05	0.81	−68.43
Δ_Day-7–Day-0_	276	−2.51	1.53	−6.99	0.98	−61.00
Δ *E **_ab_ ^3^	276	10.36	3.54	3.70	19.74	34.18
TBARS ^4^, mg MDA/kg tissue						
Day 0 TBARS	272	0.35	0.25	0.02	1.56	71.12
Day 7 TBARS	272	1.05	0.59	0.10	2.71	56.23
Δ_Day-7–Day-0_	272	0.70	0.56	−0.52	2.50	81.34

^1^ Visual discoloration was based on a percentage scoring system, where 0 indicated 0% surface discoloration and 100 indicated 100% surface discoloration ^2^ Instrumental color was measured with a handheld Minolta Chroma meter (Konica Minolta Sensing Americas, Inc., Ramsey, NJ, USA) with illuminant D_65_ and 0° viewing angle settings and an 8mm diameter circular aperture. Values reported in this table include readings on day 0, day 5, day 6, and day 7 of the display period as well as the change in the readings from day 0 to day 5, day 6, and day 7 of the display period. ^3^ Δ *E**ab = [(d-7 *L** − d-1 *L**)2 + (d-7 *a** − d-1 *a**)2 + (d-7 *b** − d-1 *b**)^2^]^0.5^. ^4^ Thiobarbituric Acid Reactive Substances (TBARS).

**Table 3 foods-10-01982-t003:** Pearson correlation coefficients (r) for composition (moisture and lipid) with rate of discoloration during a simulated retail display.

Variable	Moisture, %		Lipid, %	
Visual discoloration Δ_Day-5–Day-0_	−0.51	(<0.01)	0.54	(<0.01)
Visual discoloration Δ_Day-6–Day-0_	−0.51	(<0.01)	0.51	(<0.01)
Visual discoloration Δ_Day-7–Day-0_	−0.51	(<0.01)	0.53	(<0.01)
*L** Day 0 scores	−0.54	(<0.01)	0.46	(<0.01)
*L** Δ_Day-5–Day-0_	0.12	(0.04)	0.19	(<0.01)
*L** Δ_Day-6–Day-0_	−0.15	(0.01)	0.23	(<0.01)
*L** Δ_Day-7– Day-0_	−0.23	(<0.01)	0.32	(<0.01)
*a** Day-0 scores	−0.55	(<0.01)	0.63	(<0.01)
*a** Δ_Day-5–Day-0_	0.58	(<0.01)	−0.63	(<0.01)
*a** Δ_Day-6–Day-0_	0.63	(<0.01)	−0.68	(<0.01)
*a** Δ_Day-7–Day-0_	0.65	(<0.01)	−0.70	(<0.01)
*b** Day-0 scores	−0.66	(<0.01)	0.73	(<0.01)
*b** Δ_Day-5–Day-0_	0.32	(<0.01)	−0.40	(<0.01)
*b** Δ_Day-6–Day-0_	0.32	(<0.01)	−0.38	(<0.01)
*b** Δ_Day-7–Day-0_	0.28	(<0.01)	−0.36	(<0.01)
Δ*E**_ab_	−0.62	(<0.01)	0.68	(<0.01)

Correlation coefficient between traits is on the right. *p*-value for correlation coefficients on the left and in brackets indicate the difference from zero.

**Table 4 foods-10-01982-t004:** Pearson correlation coefficients (r) for fatty acid profile (g/100 g fat tissue) with rate of discoloration during a simulated retail display.

Variable	MUFA:SFA		PUFA:SFA		PUFA:MUFA		n-6 PUFA		n-3 PUFA		n-6:n-3 Ratio	
Visual discoloration Δ_Day-5–Day-0_	0.11	(0.06)	0.07	(0.25)	0.14	(0.02)	0.10	(0.09)	0.06	(0.30)	0.08	(0.19)
Visual discoloration Δ_Day-6–Day-0_	0.13	(0.03)	−0.03	(0.62)	0.12	(0.05)	0.13	(0.03)	0.06	(0.33)	0.08	(0.16)
Visual discoloration Δ_Day-7–Day-0_	0.12	(0.04)	0.07	(0.28)	0.15	(0.01)	0.11	(0.06)	0.07	(0.26)	0.07	(0.22)
*L** Day 0 scores	−0.05	(0.39)	0.07	(0.28)	−0.06	(0.31)	−0.02	(0.69)	−0.09	(0.13)	0.01	(0.83)
*L** Δ_Day-5–Day-0_	0.18	(<0.01)	0.07	(0.22)	0.22	(<0.01)	0.13	(0.03)	0.19	(<0.01)	−0.03	(0.58)
*L** Δ_Day-6–Day-0_	0.13	(0.03)	0.06	(0.36)	0.19	(<0.01)	0.12	(0.05)	0.08	(0.20)	0.07	(0.24)
*L** Δ_Day-7– Day-0_	0.12	(0.05)	0.09	(0.14)	0.20	(<0.01)	0.08	(0.15)	0.13	(0.03)	0.02	(0.69)
*a** Day 0 scores	0.08	(0.21)	0.24	(<0.01)	0.20	(<0.01)	0.04	(0.48)	0.11	(0.06)	0.01	(0.84)
*a** Δ_Day-5–Day-0_	−0.17	(0.01)	−0.14	(0.02)	−0.23	(<0.01)	−0.17	(<0.01)	−0.06	(0.33)	−0.13	(0.04)
*a** Δ_Day-6–Day-0_	−0.12	(0.05)	−0.15	(0.01)	−0.20	(<0.01)	−0.12	(0.05)	−0.05	(0.42)	−0.11	(0.07)
*a** Δ_Day-7–Day-0_	−0.13	(0.03)	−0.19	(<0.01)	−0.21	(0.01)	−0.12	(0.05)	−0.07	(0.22)	−0.08	(0.17)
*b** Day 0 scores	0.05	(0.39)	0.19	(<0.01)	0.16	(0.01)	0.02	(0.72)	0.10	(0.11)	−0.02	(0.77)
*b** Δ_Day-5–Day-0_	−0.13	(0.03)	−0.12	(0.04)	−0.19	(<0.01)	−0.13	(0.04)	−0.06	(0.35)	−0.06	(0.31)
*b** Δ_Day-6–Day-0_	−0.11	(0.06)	−0.12	(0.05)	−0.16	(0.01)	−0.11	(0.08)	−0.06	(0.35)	−0.03	(0.68)
*b** Δ_Day-7–Day-0_	−0.07	(0.25)	−0.11	(0.08)	−0.13	(0.03)	−0.05	(0.39)	−0.07	(0.24)	0.01	(0.85)
Δ*E**_ab_	0.14	(0.02)	0.20	(<0.01)	0.23	(<0.01)	0.13	(0.03)	0.08	(0.19)	0.08	(0.16)

Correlation coefficient between traits is on the right. *p*-value for correlation coefficients on the left and in brackets indicate the difference from zero.

**Table 5 foods-10-01982-t005:** Pearson correlation coefficients (r) for composition (moisture and lipid) with rate of lipid oxidation during a simulated retail display.

Variable	Moisture, %		Lipid, %	
Day-0 TBARS, mg MDA/kg meat	−0.15	(0.01)	0.09	(0.17)
Day-7 TBARS, mg MDA/kg meat	−0.39	(<0.01)	0.46	(<0.01)
TBARSΔ_Day-7–Day-0_	−0.33	(<0.01)	0.44	(<0.01)

Correlation coefficient between traits is on the right. *p*-value for correlation coefficients on the left and in brackets indicate the difference from zero.

**Table 6 foods-10-01982-t006:** Pearson correlation coefficients (r) for fatty acid profile (g/100 g fat tissue) with rate of lipid oxidation during a simulated retail display.

Variable	MUFA:SFA		PUFA:SFA		PUFA:MUFA		n-6 PUFA		n-3 PUFA		n-6:n-3 Ratio	
Day-0 TBARS	−0.09	(0.13)	−0.08	(0.18)	−0.10	(0.09)	−0.01	(0.88)	−0.05	(0.41)	0.02	(0.69)
Day-7 TBARS	0.13	(0.03)	0.18	(<0.01)	0.26	(<0.01)	0.12	(0.05)	0.01	(0.90)	0.14	(0.02)
TBARSΔ_Day-7–Day-0_	0.17	(<0.01)	0.22	(<0.01)	0.31	(<0.01)	0.13	(0.03)	0.03	(0.62)	0.13	(0.02)

Correlation coefficient between traits is on the right. *p*-value for correlation coefficients on the left and in brackets indicate the difference from zero.

**Table 7 foods-10-01982-t007:** Pearson correlation coefficients (r) for the rate of lipid oxidation with the rate of discoloration during a simulated retail display.

Variable	Day-0 TBARS		Day-7 TBARS		TBARSΔ_Day-7–Day-0_	
Visual discoloration Δ_Day-7–Day-0_	0.01	(0.92)	0.35	(<0.01)	0.37	(<0.01)
*L** Day 7 scores	0.12	(0.05)	0.26	(<0.01)	0.22	(<0.01)
*L** Δ_Day-7– Day-0_	0.11	(0.08)	0.44	(<0.01)	0.50	(<0.01)
*a** Day 7 scores	0.07	(0.25)	−0.26	(<0.01)	−0.24	(<0.01)
*a** Δ_Day-7–Day-0_	0.03	(0.59)	−0.55	(<0.01)	−0.55	(<0.01)
*b** Day 7 scores	0.06	(0.34)	0.39	(<0.01)	0.38	(<0.01)
*b** Δ_Day-7–Day-0_	0.06	(0.33)	−0.35	(<0.01)	−0.38	(<0.01)
Δ*E**_ab_	0.01	(0.83)	0.56	(<0.01)	0.57	(<0.01)

Correlation coefficient between traits is on the right. *p*-value for correlation coefficients on the left and in brackets indicate the difference from zero.

## Data Availability

Data can be made available upon request.

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
