# Peer review of "The Relationship between Lipid Content in Ground Beef Patties with Rate of Discoloration and Lipid Oxidation during Simulated Retail Display"

_foods, 2021, doi:10.3390/foods10091982_

Round 1
Reviewer 1 Report
This study evaluated the correlation between the fat content and in discoloration of beef patties during simulated display storage. Since discoloration is a key indicator of changes in the quality of beef patties during storage, the objective of this study is reasonable. However, there is a question as to whether the influencing factors other than the fat content were completely controlled, since samples were made from several animal populations. Moreover, since the obtained results still confirm the relationship between fat content and discoloration, it is necessary to further emphasize the strength of this study.
In the present introduction, background knowledge on discoloration and myoglobin/lipid oxidation due to oxidative quality change could be obtained, but it is insufficient to understand the research hypothesis on the relationship between lipid content and oxidative quality change. Thus, the strong description on research hypothesis should be addressed.
L77-81 Since it is not common to make patties using frozen meat, please explain the reason for using frozen meat in this study.
L83-86 Please mention how long days the frozen beef and fat were stored.
The data shown in Figures 1 and 2 is an obvious result because the fat content was controlled. I think it is unnecessary if there is nothing that we really want to mention through this data.
L135-157 Although the addition of subcutaneous fat may dominate the fatty acid composition of patties, the reason for using subcutaneous fat samples rather than using patty samples should be explained in that fat is also present in skeletal muscle.
L171 Please describe how to do visual evaluation on surface discoloration.
L175-184 Please mention the aperture size, since two measurements per patty would be insufficient to represent color characteristics.
L362 It is unreasonable to give a strong meaning based on an R-squared of 0.5 or less. Moreover, it is difficult to understand discoloration only with redness and delta E, so it would be better to add hue angle values for better expressing instrumental discoloration.
Author Response
Comments and Suggestions for Authors
This study evaluated the correlation between the fat content and in discoloration of beef patties during simulated display storage. Since discoloration is a key indicator of changes in the quality of beef patties during storage, the objective of this study is reasonable. However, there is a question as to whether the influencing factors other than the fat content were completely controlled, since samples were made from several animal populations. Moreover, since the obtained results still confirm the relationship between fat content and discoloration, it is necessary to further emphasize the strength of this study.
Thank you for these comments on our work. We have included several statements in the paper to describe these limitations that you have mentioned. Particularly on lines 248-251, where it is mentioned that “All statistical assumptions for correlation and regression modeling were met, yet it should be noted that these data are observational in nature and confounding factors related to diets the animals were fed, fresh meat quality, patty manufacture, and storage conditions during the simulated retail display periods should be recognized.”
In the present introduction, background knowledge on discoloration and myoglobin/lipid oxidation due to oxidative quality change could be obtained, but it is insufficient to understand the research hypothesis on the relationship between lipid content and oxidative quality change. Thus, the strong description on research hypothesis should be addressed.
The following hypothesis statements were added to lines 64-67:
It was hypothesized that ground beef manufactured with greater lipid content could accelerate the rate of oxidation processes, however the relationship between rate of myoglobin state change (i.e., discoloration) and lipid oxidation, while assumed to be linked, may not be strongly related.
L77-81 Since it is not common to make patties using frozen meat, please explain the reason for using frozen meat in this study.
Samples were frozen because it was not possible to manufacture all the samples on the same day, nor was it possible to collect the color stability data for the shelf-life study during the same period of time.
Ground beef patties were manufactured over nine production periods, although all samples used did experience a freeze-thaw cycle for consistency. Freezing beef will delay the rate of microbial growth, meat quality deterioration, and control the initial lipid oxidation of all the samples.
L83-86 Please mention how long days the frozen beef and fat were stored.
The following was added to lines 87-88:
Boneless inside rounds and subcutaneous fat were frozen for a period of time ranging from 17 days to 72 days.
The data shown in Figures 1 and 2 is an obvious result because the fat content was controlled. I think it is unnecessary if there is nothing that we really want to mention through this data.
We agree that this is an obvious result but feel that these figures add to our story and provide greater context to the reader.
L135-157 Although the addition of subcutaneous fat may dominate the fatty acid composition of patties, the reason for using subcutaneous fat samples rather than using patty samples should be explained in that fat is also present in skeletal muscle.
We agree that the use of subcutaneous fat for these analyses is not ideal. We unfortunately do not have an appropriate explanation for this shortcoming, and will take this into consideration in our future work.
L171 Please describe how to do visual evaluation on surface discoloration.
A reference was added to the text on line 197.
Surface discoloration (%) was evaluated by two trained panelists using the following reference images (Wang et al., 2020b).
Wang, L.M.; Huang, S.; Chalupa-Krebzdak, S.; Mejía, S.M.V.; Mandell, I.B.; Bohrer, B.M. Effects of essential oils and (or) benzoic acid in beef finishing cattle diets on the fatty acid profile and shelf life stability of ribeye steaks and ground beef. Meat Sci. 2020b, 168(10), 108195. https://doi.org/10.1016/j.meatsci.2020.108195
L175-184 Please mention the aperture size, since two measurements per patty would be insufficient to represent color characteristics.
The aperture size was 8mm. This information was added to line 190-191.
L362 It is unreasonable to give a strong meaning based on an R-squared of 0.5 or less. Moreover, it is difficult to understand discoloration only with redness and delta E, so it would be better to add hue angle values for better expressing instrumental discoloration.
Thanks for these suggestions.
1) For consistency purposes, Coefficients of determination (r2) were considered weak at r2 < 0.12, moderate at 0.13 ≤ r2 < 0.45, and strong at r2 ≥ 0.46. This is according to the following two references.
Taylor, R. Interpretation of the correlation coefficient: a basic review. J. Diagn. Med. Sonog. 1990, 6(1), 35–39. https://doi.org/10.1177%2F875647939000600106
Bohrer, B.M.; Boler, D.D. Subjective pork quality evaluation may not be indicative of instrumental pork quality measurements on a study-to-study basis. Prof. Anim. Sci. 2017, 33(5), 530–540. https://doi.org/10.15232/pas.2017-01644
2) According to recent work by Salueña et al. (2019), delta E was more sensitive to metmyoglobin changes during oxidation compared with L*, b*, and hue value, and therefore a better indicator of meat discoloration during oxidation. While hue angle can also be important, it does not incorporate L*, a*, and b*, only a* and b* and does not compare change over time. With the amount of data and variables that we have already chosen to present, we are not sure that it is feasible to add hue angle to the paper and do not feel that it does add meaningful information to the interpretation of the dataset.
Salueña, B.H.; Gamasa, C.S.; Rubial, J.M.D.; Odriozola, C.A. CIELAB color paths during meat shelf life. Meat Sci. 2019, 157, 107889. https://doi.org/10.1016/j.meatsci.2019.107889
Reviewer 2 Report
The article presents information that relates to current issues, namely the relationship between lipid content in ground beef patties with rate of discoloration and lipid oxidation during simulated retail display of selected quality parameters.
Title and Abstract are corresponding to the content of the article.
There are brief information in the introduction and are documented by literary sources. However, I recommend authors to emphasize the issue of discoloration in relation to spoilage. The authors wrote the references, but I think it would be desirable to mention this in more details in a few sentences, while there is a space in this chapter. Particularly, it relates to time and temperature as significant external factors.
Chapter Materials and Methods provides all essential informations about the experiment. But, through what I mentioned in the last sentence I think the material (Animals) should be called other than Raw Material and should be more specified. This informations could be listed on references as they presented these authors in manuscript. However, it should be written in a better way....reader does not encourage it to trace in the listed references.
The muscle names should be listed Latin and thus in the italics. (also post mortem,...).
Lines 83-86: Can authors specify why these temperatures have been set and the number of days listed here? It is related to my note to the Introduction Section. Here it could be placed reference on legislation or literature, this could be related to hygiene quality characteristic.
Lines 168-169: Type and manufacturer should be listed.
Lines 169-171: This is set by authors why? Need to be explained. Reference to a similar article? Or give the reason, it cannot be presented to readers widely.
Line 189: Better is ΔE*ab (1976) description.
Results and Discussion
The Table 2 is extensive and even if there are fundamental default measured results, it may not be completely clear for the reader.
The issue of color measurement is clear for my person, but for readers it could be some complications of presented results. May be the graphical expression should be for ΔE*ab helpful.
The results are well discussed. However, how the results and conclusion provide interesting and practically useful knowledges.
I think that for these article would be better if it will be shorter and better and more clearly described reasons and consequences of this experiment.
Author Response
Comments and Suggestions for Authors
The article presents information that relates to current issues, namely the relationship between lipid content in ground beef patties with rate of discoloration and lipid oxidation during simulated retail display of selected quality parameters.
Title and Abstract are corresponding to the content of the article.
There are brief information in the introduction and are documented by literary sources. However, I recommend authors to emphasize the issue of discoloration in relation to spoilage. The authors wrote the references, but I think it would be desirable to mention this in more details in a few sentences, while there is a space in this chapter. Particularly, it relates to time and temperature as significant external factors.
We are in complete agreement that discoloration and oxidative processes are related to spoilage, but that is unfortunately beyond the scope of this study’s objectives. We did not measure bacteria growth or other indicators of spoilage in this study, and rather focused our efforts on discoloration and lipid oxidation. We are unsure if adding such sections to the introduction would improve our message to the reader.
Chapter Materials and Methods provides all essential informations about the experiment. But, through what I mentioned in the last sentence I think the material (Animals) should be called other than Raw Material and should be more specified. This informations could be listed on references as they presented these authors in manuscript. However, it should be written in a better way....reader does not encourage it to trace in the listed references.
The title of section 2.1 was revised to “Production of animals and procurement of raw materials”
The muscle names should be listed Latin and thus in the italics. (also post mortem,...).
Revised.
Lines 83-86: Can authors specify why these temperatures have been set and the number of days listed here? It is related to my note to the Introduction Section. Here it could be placed reference on legislation or literature, this could be related to hygiene quality characteristic.
After some initial testing, it took 5 days in a refrigerated room (≤ 4°C) for samples to be completely thawed.
Lines 168-169: Type and manufacturer should be listed.
Information was added to lines 174-175.
Lines 169-171: This is set by authors why? Need to be explained. Reference to a similar article? Or give the reason, it cannot be presented to readers widely.
According to AMSA (2012), 60% of surface discoloration is considered as modest discoloration, and an appropriate level of oxidation to measure differences between samples. If all samples were 100% discolored, then researchers are unable to detect meaningful differences in other measurements such as lipid oxidation at the completion of the simulated retail display period. This information was added on lines 184-187.
Line 189: Better is ΔE*ab (1976) description.
This information was added in Line 203.
Results and Discussion
The Table 2 is extensive and even if there are fundamental default measured results, it may not be completely clear for the reader.
Thank you for these suggestions, we have included an additional footnote for this table.
The issue of color measurement is clear for my person, but for readers it could be some complications of presented results. May be the graphical expression should be for ΔE*ab helpful.
Thank you for this suggestion, yet we are unsure of what type of graphical expression you are suggesting. If possible, could you please clarify.
The results are well discussed. However, how the results and conclusion provide interesting and practically useful knowledges.
Thank you for these comments.
I think that for these article would be better if it will be shorter and better and more clearly described reasons and consequences of this experiment.
Thank you for these suggestions. We agree that this article has ended up being a longer article with more data than what we had originally anticipated. If possible, could you please provide us with specific areas of the manuscript that we should attempt to eliminate or shorten.
Reviewer 3 Report
Dear Authors
the manuscript you presented represents a good contribution to evaluate the shelf-live of meat products, however, despite the breadth of the results reported and the supporting discussions, it presents some critical issues that must be addressed. In general, attention should be paid to the numerical bibliography reported in the text and its correspondence with that of the section at the end of the manuscript. For example, line 261 returns reference 36 for the first time, skipping 35. Check the entire document. Line 87, reference is made to obtaining 276 batches using the meat of 138 carcasses. it is reasonable that for each carcass 2 batches were obtained, how do they differ? specify better. Lines 91-93, what is meant by samples with different levels of fat? specify better, or connect the sentence more appropriately with the subsequent explanatory sentence. Lines 254-256, it is reported that the total fatty acids is determined as the sum of SFA, MUFA and PUFA, in reality they should be quantified on the basis of an equation that takes into account the internal standard used, the area of the peak of the internal standard and the total area of the identified peaks. Table 1, check the values. For example, the average value of the PUFA totals differs from the sum of n-6 PUFA and n-3 PUFA, and is also lower. Table 2, in the face of a consistent variation of visual discoloration, a * and b *, it seems too contrasting that the average values of L * remain almost constant. Furthermore, the mean value of the difference in TBARS at 7 and 0 days seems excessively high compared to the mean values of TBARS at 7 and 0 days. Finally, but perhaps most importantly, the results show the fatty acid profile of the subcutaneous fat used to prepare the mixtures, I understand that the characteristics of the meat are reported in Wang et al. (2020b) and Dorleku et al. (2021), however, in the whole manuscript there is no reverence for the fatty acid profile of the mixtures of meat and fat, rather than the fatty acid profile of the mixtures is correlated with the visual, colorimetric and oxidative parameters.
Best regards
Author Response
Comments and Suggestions for Authors
foods-1326003-peer-review-v1
The relationship between lipid content in ground beef patties with rate of discoloration and lipid oxidation during simulated retail display. This article represents a significant contribution to scientific knowledge about the relationships between lipid content, lipid oxidation and the discoloration rate of ground beef in a simulated retail exhibition. In addition, the results found are helpful to transpose to an actual commercial situation.
Thank you for these comments.
The article is well organized and written clearly. Despite this, some improvements can be made to make the text clearer. The text addresses the subject with scientific accuracy. Tables and figures are relevant. In Tables 3, 4, 5, 6 and 7, if the P-value is placed next to the correlation value, it will decrease the size of the tables.
Revised.
Regarding figures, it is desirable to improve their quality. The numbers and letters are pixelated. The methods are clearly described, which will allow a perfect understanding by other researchers. The results are well presented and discussed with the existing knowledge on the subject. The results support the conclusions.
Thank you for these comments. Figures were revised.
If it is not too much work, please remove the decimals in values of the axes in all Figures. Please maintain consistency.
Figures were revised.
L56 in a concurrent manner and “change with” in a concurrent manner and
Revised.
L59 were to first evaluate the relationship “change with” were first to evaluate the relationship
Revised.
L139 subcutaneous fat approximately “change with” subcutaneous fat of approximately
Revised.
L143 Extraction efficiency of “change with” The extraction efficiency of
Revised.
181-183 Visual discoloration (%) was evaluated by two trained panelists on each day of the shelf-life study using Meat Color Measurement Guidelines outlined by AMSA [25]. “change with” Two trained panelists evaluated visual discoloration (%) on each day of the shelf-life study using Meat Color Measurement Guidelines outlined by AMSA [25].
Revised.
L205 were weighed immediately following thawing for “change with” were weighed the following thawing immediately for
Revised.
L215-218 Samples were corrected using recovery rate percentages captured using spiked samples which consisted of adding 1 mL of 0.2 mg/mL of BHT, 12 mL of TEP, and 32 mL of TCA/H3PO4. “change with” Samples were corrected using recovery rate percentages captured using spiked samples, which added 1 mL of 0.2 mg/mL of BHT, 12 mL of TEP, and 32 mL of TCA/H3PO4.
Revised.
L219 test samples. All samples “change with” test samples. Therefore, all samples
Revised.
253 for fatty acid “change with” for the fatty acid
Revised.
L295 Moderate positive “change with” A moderate positive
Revised.
L298 in the light scattering “change with” in the light-scattering
Revised.
L305-307 High ultimate pH could improve water holding capacity, and therefore increase the content of water-soluble myoglobin in the meat system. “change with” High ultimate pH could improve water holding capacity and increase the content of water-soluble myoglobin in the meat system.
Revised.
L312 related with a less “change with” red related to a less red
Revised.
LO324-326 Likewise, Cooper et al. [48] observed ground beef formulated with 25% fat demonstrated lower a* values compared with counterpart samples formulated with 5% fat during 7 days of retail display. “change with” Likewise, Cooper et al. [48] observed that ground beef formulated with 25% fat demonstrated lower a* values than counterpart samples formulated with 5% fat during 7 days of retail display.
Revised.
L341-342 In general, increased lipid content not only favored an increase in initial yellowness, but also slowed down the change in yellowness during retail display. “change with” In general, high lipid content favored an increase in initial yellowness and slowed down the change in yellowness during retail display.
Revised.
L400 PUFA and redness “change with” PUFA and the redness
Revised.
L420 display, however, it “change with” display; however, it
Revised.
L430 In contrast to the results of the current study, Cooper et al. [48] reported “change with” In contrast to the current study results, Cooper et al. [48] reported
Revised.
L447 Thus, it is plausible that not only the fatty acid composition, but also the total fat content is an important factor in the development of lipid oxidation “change with” Thus, it is plausible that the fatty acid composition and the total fat content are important factors in the development of lipid oxidation.
Revised.
L454 difficult to reach an understanding of “change with” the difficult to understand the
Revised.
L460 discussion were calculated “change with” discussion was calculated
Revised.
L503-505 Therefore, the question of whether fatty acid composition or lipid content has a greater impact in lipid oxidation may be dependent on the form of meat products. “change with” Therefore, whether fatty acid composition or lipid content has a greater impact on lipid oxidation may depend on the form of meat products.
Revised.
L567 lipid oxidation, since “change with” lipid oxidation since
Revised.
L573 beef which “change with” beef, which
Revised.
L575 agreement to the findings “change with” agreement with the findings
Revised.
Reviewer 4 Report
foods-1326003-peer-review-v1
The relationship between lipid content in ground beef patties with rate of discoloration and lipid oxidation during simulated retail display
This article represents a significant contribution to scientific knowledge about the relationships between lipid content, lipid oxidation and the discoloration rate of ground beef in a simulated retail exhibition. In addition, the results found are helpful to transpose to an actual commercial situation.
The article is well organized and written clearly. Despite this, some improvements can be made to make the text clearer. The text addresses the subject with scientific accuracy. Tables and figures are relevant. In Tables 3, 4, 5, 6 and 7, if the P-value is placed next to the correlation value, it will decrease the size of the tables.
Regarding figures, it is desirable to improve their quality. The numbers and letters are pixelated. The methods are clearly described, which will allow a perfect understanding by other researchers. The results are well presented and discussed with the existing knowledge on the subject. The results support the conclusions.
Some detailed comments below:
L56 in a concurrent manner and “change with” in a concurrent manner and
L59 were to first evaluate the relationship “change with” were first to evaluate the relationship
If it is not too much work, please remove the decimals in values of the axes in all Figures. Please maintain consistency.
L139 subcutaneous fat approximately “change with” subcutaneous fat of approximately
L143 Extraction efficiency of “change with” The extraction efficiency of
181-183 Visual discoloration (%) was evaluated by two trained panelists on each day of the shelf-life study using Meat Color Measurement Guidelines outlined by AMSA [25]. “change with” Two trained panelists evaluated visual discoloration (%) on each day of the shelf-life study using Meat Color Measurement Guidelines outlined by AMSA [25].
L205 were weighed immediately following thawing for “change with” were weighed the following thawing immediately for
L215-218 Samples were corrected using recovery rate percentages captured using spiked samples which consisted of adding 1 mL of 0.2 mg/mL of BHT, 12 mL of TEP, and 32 mL of TCA/H3PO4. “change with” Samples were corrected using recovery rate percentages captured using spiked samples, which added 1 mL of 0.2 mg/mL of BHT, 12 mL of TEP, and 32 mL of TCA/H3PO4.
L219 test samples. All samples “change with” test samples. Therefore, all samples
253 for fatty acid “change with” for the fatty acid
L295 Moderate positive “change with” A moderate positive
L298 in the light scattering “change with” in the light-scattering
L305-307 High ultimate pH could improve water holding capacity, and therefore increase the content of water-soluble myoglobin in the meat system. “change with” High ultimate pH could improve water holding capacity and increase the content of water-soluble myoglobin in the meat system.
L312 related with a less “change with” red related to a less red
LO324-326 Likewise, Cooper et al. [48] observed ground beef formulated with 25% fat demonstrated lower a* values compared with counterpart samples formulated with 5% fat during 7 days of retail display. “change with” Likewise, Cooper et al. [48] observed that ground beef formulated with 25% fat demonstrated lower a* values than counterpart samples formulated with 5% fat during 7 days of retail display.
L341-342 In general, increased lipid content not only favored an increase in initial yellowness, but also slowed down the change in yellowness during retail display. “change with” In general, high lipid content favored an increase in initial yellowness and slowed down the change in yellowness during retail display.
L400 PUFA and redness “change with” PUFA and the redness
L420 display, however, it “change with” display; however, it
L430 In contrast to the results of the current study, Cooper et al. [48] reported “change with” In contrast to the current study results, Cooper et al. [48] reported
L447 Thus, it is plausible that not only the fatty acid composition, but also the total fat content is an important factor in the development of lipid oxidation “change with” Thus, it is plausible that the fatty acid composition and the total fat content are important factors in the development of lipid oxidation.
L454 difficult to reach an understanding of “change with” the difficult to understand the
L460 discussion were calculated “change with” discussion was calculated
L503-505 Therefore, the question of whether fatty acid composition or lipid content has a greater impact in lipid oxidation may be dependent on the form of meat products. “change with” Therefore, whether fatty acid composition or lipid content has a greater impact on lipid oxidation may depend on the form of meat products.
L567 lipid oxidation, since “change with” lipid oxidation since
L573 beef which “change with” beef, which
L575 agreement to the findings “change with” agreement with the findings
Author Response
Comments and Suggestions for Authors
Dear Authors
the manuscript you presented represents a good contribution to evaluate the shelf-live of meat products, however, despite the breadth of the results reported and the supporting discussions, it presents some critical issues that must be addressed. In general, attention should be paid to the numerical bibliography reported in the text and its correspondence with that of the section at the end of the manuscript.
For example, line 261 returns reference 36 for the first time, skipping 35. Check the entire document.
Revised.
Line 87, reference is made to obtaining 276 batches using the meat of 138 carcasses. it is reasonable that for each carcass 2 batches were obtained, how do they differ? specify better.
Two batches of ground beef with different lean:fat levels were produced for each carcass. This is further described on line 101-109.
Lines 91-93, what is meant by samples with different levels of fat? specify better, or connect the sentence more appropriately with the subsequent explanatory sentence.
This is described in figure 2 as well with the text from lines 101-109.
Lines 254-256, it is reported that the total fatty acids is determined as the sum of SFA, MUFA and PUFA, in reality they should be quantified on the basis of an equation that takes into account the internal standard used, the area of the peak of the internal standard and the total area of the identified peaks.
Revised wording was used here. Lines 266-267. I believe what we are getting to are the same quantification, correct? Please correct us if we are not following your line of thinking.
Table 1, check the values. For example, the average value of the PUFA totals differs from the sum of n-6 PUFA and n-3 PUFA, and is also lower.
Thank you very much for noticing our errors. We have taken a deep dive into the data and have corrected and adjusted accordingly.
Table 2, in the face of a consistent variation of visual discoloration, a * and b *, it seems too contrasting that the average values of L * remain almost constant. Furthermore, the mean value of the difference in TBARS at 7 and 0 days seems excessively high compared to the mean values of TBARS at 7 and 0 days.
Thank you very much for noticing our errors. We have taken a deep dive into the data and have corrected and adjusted accordingly.
Finally, but perhaps most importantly, the results show the fatty acid profile of the subcutaneous fat used to prepare the mixtures, I understand that the characteristics of the meat are reported in Wang et al. (2020b) and Dorleku et al. (2021), however, in the whole manuscript there is no reverence for the fatty acid profile of the mixtures of meat and fat, rather than the fatty acid profile of the mixtures is correlated with the visual, colorimetric and oxidative parameters.
We agree that the use of subcutaneous fat for these analyses is not ideal. We unfortunately do not have an appropriate explanation for this shortcoming, and will take this into consideration in our future work.
Round 2
Reviewer 1 Report
The pointed issues have been handled appropriately.
Reviewer 3 Report
Dear Authors
the manuscript can be approved in the modified form
Best regards